# Development of a Comprehensive Quality Evaluation System for Foxtail Millet from Different Ecological Regions

**DOI:** 10.3390/foods12132545

**Published:** 2023-06-29

**Authors:** Liguang Zhang, Ke Ma, Xiatong Zhao, Zhong Li, Xin Zhang, Weidong Li, Ru Meng, Boyu Lu, Xiangyang Yuan

**Affiliations:** 1College of Agriculture, Shanxi Agricultural University, Taiyuan 030801, China; zhangliguang1982@163.com (L.Z.); 15535484140@163.com (K.M.); zhaoxiat@163.com (X.Z.); lz1711161715@163.com (Z.L.); zx1150472698@163.com (X.Z.); li15110671274@163.com (W.L.); s20212192@stu.sxau.edu.cn (R.M.); 18735424089@163.com (B.L.); 2College of Agriculture, China Agricultural University, Beijing 100089, China

**Keywords:** foxtail millet, quality evaluation system, Shanxi province, amino acid pattern

## Abstract

Foxtail millet (*Setaria italica* L.) is a critical grain with high nutritional value and the potential for increased production in arid and semiarid regions. The foxtail millet value chain can be upgraded only by ensuring its comprehensive quality. Thus, samples were collected from different production areas in Shanxi province, China, and compared in terms of quality traits. We established a quality evaluation system utilizing multivariate statistical analysis. The results showed that the appearance, nutritional content, and culinary value of foxtail millet produced in different ecological regions varied substantially. Different values of amino acids (DVAACs), alkali digestion values (ADVs), and total flavone content (TFC) had the highest coefficients of variation (CVs) of 50.30%, 39.75%, and 35.39%, respectively. Based on this, a comprehensive quality evaluation system for foxtail millet was established, and the quality of foxtail millet produced in the five production areas was ranked in order from highest to lowest: Dingxiang > Zezhou > Qinxian > Xingxian > Yuci. In conclusion, the ecological conditions of Xinding Basin are favorable for ensuring the comprehensive quality of foxtail millet.

## 1. Introduction

Foxtail millet (*Setaria italica* L.) belongs to the *Setaria* genus of the Poaceae grass family and is a key crop species. It is an emerging model plant for C_4_ grasses [1,2]. Foxtail millet is not only adapted to drought and barren environments but also has the advantage of high production efficiency and low resource consumption. It is suitable for water conservation and sustainable agriculture in arid and semiarid regions and has been recognized as a critical strategic reserve crop for future complex climate environments [3,4]. Hulled foxtail millet contains high levels of resistant starch, protein, amino acids, free fatty acids, dietary fiber, vitamins, antioxidants, and minerals [5,6,7]. Dietary fiber, phytochemicals, and bran lipids have also been widely studied as potential ingredients in functional foods [8,9]. In addition, several studies have reported that foxtail millet possesses a series of pharmacological benefits, including the treatment of type 2 diabetes and cardiovascular diseases, enhancement of immunity, hypolipidemic effects, and promotion of digestion [10,11,12,13]. Hence, consuming foxtail millet may help improve and maintain health and provide a diverse diet to the consumers. Owing to its medicinal and culinary properties, consumer demand for foxtail millet is rising steadily, indicating an increasing opportunity for the foxtail millet industry to develop it as a new and healthy food source [14].

Upgrading the foxtail millet value chain is propelled by ensuring comprehensive quality [15]. The comprehensive quality of foxtail millet varies depending on the audience (producers, processors, sellers, and consumers), and it can be roughly divided into milling, appearance, nutritional, functional, culinary, hygiene, and storage qualities [16]. Appearance, nutrition, functionality, and cooking characteristics are crucial factors for consumers. Appearance quality includes the shape and color of foxtail millet grains; the dehulled grains with a spherical shape and bright yellow color are regarded as having outstanding appearance quality [17]. Starch, protein, fat, amino acids, dietary fiber, and minerals are components of nutritional quality. Previous studies have shown that foxtail millet has a higher nutritional value than other major cereal grains, with double the protein content, four-fold mineral and fat content, and triple the calcium content of rice [18]. In addition, foxtail millet is rich in RS-5-type resistant starch; the amylose–fat complex formed between starch and fat during millet cooking can delay the digestion and absorption rates of glucose in the small intestine [11,19]. Functional quality refers to the components of foxtail millet grains that have immune functions and antioxidant capacity, including phytic acid, polysaccharides, flavones, folic acid, polyphenols, and carotenoid [20]. The potential health benefits of these substances have been highlighted to develop foxtail millet as a dietary supplement [9]. The culinary quality of foxtail millet is related to the ease of stewing the grains and the flavor of the resulting porridge [17]. Consumers prefer foxtail millet, which has high palatability and is easy to cook. Notably, a single evaluation from the perspective of appearance, nutritional value, and culinary quality is not representative; therefore, it is necessary to establish a comprehensive evaluation system for foxtail millet. The relationship between different quality indicators can be clarified using multivariate statistical analyses, such as cluster analysis, correlation analysis, and principal component analysis. After classifying and simplifying the indicators, a comprehensive evaluation may be carried out [21].

Foxtail millet is primarily cultivated in Shanxi province, China, with cultivation taking place throughout the majority of the province [22]. The foxtail millet planting area is 200,000–230,000 ha annually, spanning six latitudes, and includes the spring sowing early-maturing areas, spring-sowing mid-late-maturing areas, and summer sowing areas. Our previous research identified the effects of climate and soil factors on the appearance, cooking, and eating quality of foxtail millet; however, a suitable evaluation method for the comprehensive quality of foxtail millet does not exist, and the performance of comprehensive quality in different ecological regions is unclear [23]. In this study, we selected five main foxtail millet production areas in Shanxi province belonging to different ecological regions with different climate and soil types. We cultivated the high-quality foxtail millet variety, Jingu 21, in the major production areas and analyzed the quality traits, including appearance, nutritional value, functional value, and culinary quality. The aims of the present investigation were to: (i) explore the comprehensive quality differences of foxtail millet from different ecological habitats and (ii) establish a comprehensive and conventional quality evaluation system for foxtail millet produced in different ecological regions.

## 2. Materials and Methods

### 2.1. Site Description

The experimental site was located in the hilly and mountainous areas of the semiarid and arid regions in Shanxi, China. Five production areas were selected: Dingxiang (764 m; 112.92° E, 38.58° N; Xinding Basin in Northwestern Shanxi; light brown soil), Xingxian (1022 m; 110.19° E, 38.19° N; Lvliang Mountain; gray cinnamonic soil), Yuci (1179 m; 112.86° E, 37.82° N; Jinzhong Basin in central Shanxi; cinnamon and meadow soil), Qinxian (1036 m; 112.66° E, 36.73° N; Taihang Mountain in southeast Shanxi; loess, brown soil, and meadow soil), and Zezhou (910 m; 113.02° E, 35.58° N; Zhongtiao Mountain in southeast Shanxi; leached brown soil, red clay, and brown soil). The production areas included spring sowing early-maturing areas, spring sowing mid–late-maturing areas, and summer sowing areas. In addition, the five sites belong to different ecological regions and have large differences in climate and soil conditions, which had strong representation in the regional scope. The initial soil chemical properties are listed in Appendix A.

### 2.2. Material

Jingu 21, a representative elite foxtail millet cultivar with high yield and high quality, was used in the experiment. The seeds were obtained from the Millet Research Institute of Shanxi Agriculture University (Changzhi, China).

Ethyl alcohol, Phenol, Thymol blue, Folin phenol, and Rutin were purchased from Beijing Solarbio Science & TechnologyCo., Ltd., Beijing, China. NaNO_2_, Al(NO_3_)_3_, KOH, NaOH, and sulfuric acid were purchased from Guoyao Group Chemical Reagent Co., Ltd., Beijing, China.

### 2.3. Experimental Design

A three-replicate randomized complete block design was used for field experiments. The ground was prepared before sowing, and all the fertilizers were applied at once as a base fertilizer. The foxtail millet was sown between May and June and harvested between September and October in the five production areas, with a density of 3.75 × 10^5^ seeds per ha and a spacing of 50 cm. No irrigation was performed during the entire growing period of each foxtail millet. Weeding was carried out at the seedling, jointing, and booting stages. Diseases and insect pests were well managed at all experimental sites.

### 2.4. Measurements

Prior to quality determination, foxtail millet grains were air-dried, stored at room temperature for three months, shelled using a paddy huller (JLGJ4.5, Taizhou Food Instrument Factory, Taizhou, China), pulverized into flour using ultra-centrifugal grinding (MGS-1000, Linqu Metech Automatic Control Equipment Technology Co., Ltd., Linqu, China), and then sieved using a 100-mesh screen. The shelled grains (Appendix A) and screened flour were sealed in ziplock bags and stored at −20 °C.

#### 2.4.1. Measurement of Appearance Quality

The 1000-grain weight (KGW) of the shelled foxtail millet grains was measured using a 10,000 g capacity analytical balance (Mettler-Toledo, LLC., Shanghai, China). The length of 100 grains (L) was measured using a ruler, and the average diameter of one grain (DG) was calculated according to the following equation:(1)$$DG(mm)=\frac{L}{100}$$

The color parameters (L*: brightness, a*: red+ and green-, b*: yellow+ and blue-) of shelled grains were determined using a colorimeter (X-Rite VS450, Big Rapids, MI, USA) [18]. The color contribution index (CCI) was calculated according to the following equation [24].
(2)$$CCI=\frac{1000\times a*}{L*\times b*}$$

#### 2.4.2. Measurement of Culinary Quality

The alkali digestion value (ADV) was determined according to the method described by Ning et al. [21]. Twenty shelled foxtail millet grains of uniform size were placed in a Petri dish, and 10 mL of 1.7% KOH was pipetted into the dish until the grains were completely submerged. The samples were incubated in a 30 °C thermostat incubator for 6 h, and grain decomposition was observed. As shown in Appendix A, the grade of decomposition (*G*) was classified on a scale of 1–7, and the number of grains at each grade (*N*) was recorded. The ADV was calculated according to the following equation:(3)$$ADV=\frac{\sum_{1}^{i}G_{i}\times N_{i}}{20}$$

Gel consistency (GC) was estimated using the method described by Ning et al. [21]. Foxtail millet flour (100 mg) was weighed into 13 mm × 100 mm tubes and then 200 µL ethyl alcohol (95%), containing 0.025% thymol blue, and 2.5 mL 0.15 mol L^−1^ KOH were added to each tube. The suspension was mixed using a vortex mixer (MX-S, DLAB Scientific, Beijing, China) and placed in a vigorously boiling water bath for 8 min. After the tubes were removed from the water bath, they were maintained at room temperature for 5 min and cooled in ice water for 20 min. The tubes were then placed horizontally on a light box on top of graphing paper, and the gel migration distance was measured after 1 h.

The water solubility index (WSI) and water absorption index (WAI) were determined using the AOAC method [25]. Exactly 1.5 g (*W*_0_) of foxtail millet flour was thoroughly mixed with 18 mL of distilled water in a centrifuge tube (*W*_1_). After shaking in a water bath at 30 °C for 30 min, the mixture was centrifuged at 4500 rpm for 15 min. Subsequently, the supernatant was poured into an evaporation dish (*W*_2_) and dried at 105 °C until reaching the constant weight (*W*_3_). The mass of the sediment (*W*_4_) was measured, and the WSI and WAI were calculated using the following equations:(4)$$WSI(\%)=\frac{W_{3}-W_{2}}{W_{0}}\times 100$$
(5)$$WAI(\%)=\frac{W_{4}-W_{1}}{W_{0}}\times 100$$

#### 2.4.3. Measurement of Nutritional Quality and Amino Acid Pattern

Moisture content (MC), amylose content (ACC), crude fat content (CFC), crude protein content (CPC), and amino acid patterns were measured using a near-infrared spectrum analyzer (NIRS^TM^DS2500, FOSS, Hillerød, Denmark). Flavor amino acids can be divided into three categories: umami (UAAC), sweet (SAAC), and bitter (BAAC). Umami amino acids include Asp and Glu; sweet amino acids include Ala, Pro, Thr, Gly, and Ser; and bitter amino acids include Leu, Phe, Val, Ile, Lys, Tyr, Met, His, and Arg [26,27]. Different values of amino acids (DVAACs) were calculated using the following equation:(6)$$DVAAC\left(\%\right)=UAAC+SAAC-BAAC$$

The quality model was provided by the instrument manufacturer, taking 860 foxtail millet samples with different genetic backgrounds and different production areas as the research object for modeling. The spectral information was collected by a near-infrared spectrometer. The determination model of foxtail millet-nutritional quality and amino acid pattern was established by spectral preprocessing methods such as standard normal variation (SNV), convolution smoothing (Detrend), and partial least squares (PLSR) modeling [28].

Total polyphenol (TPC), flavonoid (TFC), polysaccharide (PC), and yellow pigment contents (YPC) were determined according to the method described by Ma et al. [23]. The total polyphenols were extracted ultrasonically using 73% ethanol as solvent, and the standard curve was constructed with a gallic acid standard. The TFC in foxtail millet powder was determined by the Folin phenol method. The total polyphenols were extracted by ultrasound with 60% ethanol as solvent, and a standard curve was obtained by using rutin as a standard sample. TFC in foxtail millet flour was determined by sodium nitrite and the aluminum nitrate colorimetric method. Polysaccharides were ultrasonically extracted with 0.8 mol/L NaOH as solvent, and the PC in foxtail millet flour was measured by the phenol-sulfuric acid method using a glucose standard curve. YPC was determined by the extraction of n-butanol.

### 2.5. Statistical Analysis

Statistical analysis was performed with DPS 7.5 and plotted using Origin 2021. All experiments were performed at least in triplicate, and results are presented as mean ± standard deviation. Data were subjected to one-way analysis of variance and Duncan’s multiple comparison analysis. For cluster analysis, the Euclidean distance was used as the distance between the grain quality indicators of the tested foxtail millet, and the Pearson method was used to establish a clustering tree diagram. Correlation analysis was conducted using the Pearson method, and results with corresponding probability values of *p* < 0.05, *p* < 0.01, and *p* < 0.001 represented different degrees of correlation. For principal component analysis (PCA), foxtail millet quality traits were normalized to form a correlation matrix. A correlation matrix was used to determine the eigenvalues and relative contribution rates, and the factor scores of the principal components of each production area were calculated [3].

## 3. Results

### 3.1. Regional Diversity of Quality Traits

#### 3.1.1. Appearance Quality

In general, foxtail millet with round and full grains, as well as bright and yellow colors, is favored by consumers [7,18]. Millet’s appearance is a reflection of the grain quality, including grain shape and color [17]. There were significant differences in the quality and appearance of foxtail millet grains from different production areas (Figure 1A–C, Appendix A). The KGW and DG ranged from 2.6 g to 2.8 g and 1.56 mm to 1.70 mm, respectively. The CV of a* (6.27) was the highest, followed by that of CCI, b*, and L*. Li et al. [29] found that the grain length and width of rice (*Oryza sativa*) are significantly different in different rice cultivation areas, and Sun et al. [3] suggested that there are significant differences in the color of foxtail millet from different origins. As shown in Figure 1A–C, foxtail millet produced in Dingxiang had the highest KGW, DG, a*, b*, and CCI values; however, the L* value was the lowest in this region and was significantly lower than that in Qinxian, Zezhou, and Xingxian by 1.43%, 3.77%, and 3.53%, respectively. Yuci had the lowest KGW, a*, and b* values, and Zezhou had the lowest DG and CCI values, which were significantly lower than those of Qinxian, Dingxiang, Xingxian, and Yuci by 7.14%, 8.24%, 3.11%, and 2.50% for DG and 5.26%, 11.68%, 3.82%, and 5.50% for CCI, respectively. Higher KGW, DG, L*, and b* values are considered essential attributes of elite foxtail millet [3,17]. The foxtail millet grains produced in Dingxiang were full and golden in color, whereas the appearance of foxtail millet produced in Yuci and Zezhou was poor.

#### 3.1.2. Nutritional Quality

The CV of nutritional quality traits ranged from large to small in the order of TFC > TPC > PC > CPC > CFC > YPC > ACC > MC (Figure 1, Appendix A). Amylose, crude fat, and crude protein are the basic nutrients in foxtail millet, and there were considerable differences in the amount of these nutrients between the different production areas (Figure 1D–G). The CFC of foxtail millet produced in Dingxiang was the highest, whereas the CPC was the lowest. Zezhou had the highest ACC and lowest CFC, which were significantly higher or lower, respectively, than those of Qinxian, Dingxiang, Xingxian, and Yuci. Carotenoids are the primary components of yellow pigments, and there is a close correlation between foxtail millet coloration and carotenoid content [14]. In addition, carotenoids have a variety of functions, including eye protection, antioxidation, anticancer, and prevention effects [30]. The YPC ranged from 54.59 to 62.97 µg/g in five production areas. Foxtail millet produced in Dingxiang had the highest YPC, which was significantly higher than those of Qinxian, Zezhou, Xingxian, and Yuci by 11.55%, 4.27%, 7.37%, and 15.35%, respectively. Ning et al. [21] collected foxtail millet (Changnong 35) from five locations and found that the YPC was significantly different in each location. Natural polysaccharides play an important role in human health and have important medicinal value in immune enhancement, antioxidation, liver protection, and anti-diabetic and anti-tumor activities [31]. In our study, the PC of millet produced in Dingxiang was the highest and significantly higher than that of the other four areas. Polyphenols and flavones are important antioxidants that can reduce the incidence of cardiovascular diseases and type 2 diabetes as well as decrease body weight and serum low-density lipoprotein cholesterol levels [32,33,34]. The TFC and TPC were 45.18–103.21 mg/100 g and 51.35–78.03 mg/100 g, respectively. The TFC was highest in Qinxian and significantly higher than that in Zezhou, Dingxiang, Xingxian, and Yuci by 51.78%, 116.87%, 128.44%, and 59.87%, respectively. Foxtail millet produced in Qinxian also had the highest TPC, which was significantly higher than that of Dingxiang and Yuci (51.96% and 46.78%, respectively). In previous studies, phenolic acid and flavonoid accumulation in proso (*Panicum miliaceum*) and foxtail millets varied considerably between varieties [18,20]. Guo et al. [35] found that Tartary buckwheat (*Fagopyrum tatiaricum*) grown at different locations had different free and bound phenolic contents and antioxidant properties, and we found similar results when we tested foxtail millet at different locations.

#### 3.1.3. Culinary Quality

Measurements of ADV, GC, and WSI showed significant differences among the five production areas (Figure 1H,I). The ADV reflects the gelatinization temperature, with values of 1–3 representing a high gelatinization temperature (>75 °C), 4–5 a moderate temperature (70–74 °C), and 6–7 a low temperature (<69 °C). Foxtail millet produced in Qinxian and Yuci had moderate gelatinization temperatures, whereas Zezhou, Dingxiang, and Xingxian millet had high gelatinization temperatures. Foxtail millet with high ADV has a lower gelatinization temperature and is easy to cook [21]. The ADV ranged from 1.17 to 4.60, and Yuci had the highest ADV, which was significantly higher than that of Zezhou, Dingxiang, and Xingxian. Ning et al. [21] found a lower ADV of millet grown in Yuci than was found in this study, likely due to differences in characteristics and interannual environments. The GC of crops determines whether they are soft or firm when cooked [36]. Our results suggest that the GC of foxtail millet in Dingxiang was the highest, indicating that the foxtail millet of Dingxiang was soft and sticky when cooked. The WAI of foxtail millet did not differ significantly among the different production areas. Dingxiang had the lowest WSI, which was significantly lower than those of Qinxian and Xingxian by 70.55% and 35.27%, respectively. Verma et al. [7] found that the WSI of foxtail millet was significantly higher than those of barnyard millet (*Echinochloa* species) and rice. The highest CV was calculated for ADV (39.75%) followed by WSI (23.96%), GC (5.66%), and WAI (1.62%) (Appendix A).

#### 3.1.4. Amino Acid Pattern

Figure 2A shows the amino acid compositions of the Jingu 21 samples from different production areas. Among the 17 amino acids in foxtail millet, Glu content was the highest, and Cys content was the lowest. The amino acid content of the different production areas was significantly different. Foxtail millet flour has Lys as a first limiting amino acid, followed by Trp, Met, and Cys [6]. The Lys, Met, and Cys content ranged from 0.23% to 0.25%, 0.43% to 0.52%, and 0.12% to 0.18%, respectively. Amino acids were classified as essential (EAAs) or nonessential (NEAAs) [37], and the lines of each group were pentagonal in the radar chart, indicating that the compositions of EAA and NEAA in the different production areas were relatively similar (Figure 2B). The total EAA content of foxtail millet ranges from 0.39% to 0.41% in the five production areas, according to the reference pattern provided by WHO/FAO [36]. The EAA, NEAA, and total amino acid contents of foxtail millet from Dingxiang were the lowest. Our results showed that foxtail millet had the highest bitter amino acid content, followed by umami and sweet (Figure 2C). The flavor amino acid content was significantly different among different areas. The bitter amino acid content was the highest in Zezhou, which was significantly higher than that in Dingxiang and Xingxian by 9.91% and 5.21%, respectively. Yuci had the highest umami and sweet amino acid content, as well as different values of amino acids, whereas Dingxiang had the lowest. Therefore, foxtail millet produced in Qinxian, Zezhou, Xingxian, and Yuci may be more popular among consumers owing to the flavor profile.

### 3.2. Cluster Analysis

Quality indicators with strong correlations and similarities can be classified into one category using hierarchical clustering [38,39]. We conducted the cluster analysis of 22 quality parameters and found that they were clustered into six groups at the average distance of 0.6 (Figure 3A). Group I included KGW, MC, a*, GC, PC, b*, and YPC, which are mainly related to appearance, cooking, and eating characteristics. Group II comprised DG, CCI, CFC, and ADV, whereas group III comprised L* and WSI. Group IV included TPC, TFC, and WAI, which were characterized by their antioxidant properties. Group VI consisted of CPC, SAAC, UAAC, BAAC, and DVAAC, all of which are related to proteins and amino acids. The other trait (ACC) fell in Group V. Additionally, GC, CFC, L*, TPC, ACC, and CPC were the most representative indicators among the six groups (Appendix A).

### 3.3. Correlation Analysis

Analysis of the correlations between the 22 quality indicators revealed significant correlations between the factors (Figure 3B). We determined that KGW was positively (*p* ≤ 0.05) correlated with a*, MC, PC, and GC, whereas it was negatively correlated with CPC (*p* ≤ 0.01), SAAC (*p* ≤ 0.05), and BAAC (*p* ≤ 0.05). In addition, b* was significantly positively correlated with a* (*p* ≤ 0.05), YPC (*p* ≤ 0.01), and GC (*p* ≤ 0.05) and negatively correlated with SAAC (*p* ≤ 0.05) and DVAAC (*p* ≤ 0.01). Therefore, appearance was strongly correlated with nutritional quality, cooking quality, and amino acid composition. Furthermore, CPC was positively correlated with UAAC (*p* ≤ 0.05), SAAC (*p* ≤ 0.01), and BAAC (*p* ≤ 0.05) but negatively correlated with KGW (*p* ≤ 0.01), a* (*p* ≤ 0.01), MC (*p* ≤ 0.05), PC (*p* ≤ 0.05), and GC (*p* ≤ 0.01). Therefore, proteins are not directly associated with appearance or cooking quality [40]. Additionally, YPC was positively correlated with a* (*p* ≤ 0.05), b* (*p* ≤ 0.01), and GC (*p* ≤ 0.05), whereas it was negatively (*p* ≤ 0.05) correlated with SAAC and DVAAC. As for cooking and eating quality, GC was positively correlated with KGW (*p* ≤ 0.05), a* (*p* ≤ 0.001), b* (*p* ≤ 0.05), MC (*p* ≤ 0.05), YPC (*p* ≤ 0.05), and PC (*p* ≤ 0.05), whereas it was negatively correlated with CPC (*p* ≤ 0.01), UAAC (*p* ≤ 0.05), SAAC (*p* ≤ 0.001), and DVAAC (*p* ≤ 0.05).

### 3.4. Principal Component Analysis

The method of PCA enables independent factors to be extracted from a large set of intercorrelated variables. This method can also retain trends and patterns while simplifying complex data [39]. In the PCA, there were four eigenvalues greater than one, and their contribution rates were 56.147%, 24.623%, 12.282%, and 6.948% (Table 1). F_1_ was positively correlated with KGW, DG, a*, b*, CCI, MC, CFC, YPC, PC, and GC and negatively correlated with CPC, UAAC, SAAC, BAAC, and DVAAC. The results showed that when F_1_ was large, the appearance, nutritional, cooking, and eating qualities were higher, while the amino acid pattern decreased, and other quality characteristics remained unchanged. Similarly, F_2_ was positively correlated with L* and ACC but negatively correlated with DG, CCI, CFC, and ADV. Furthermore, TFC was the highest eigenvector corresponding to F_3_, followed by TPC and WAI, which indicated that when F_3_ was large, the antioxidant content was high. Moreover, WSI and L* negatively correlated with F_4_, whereas UACC positively correlated with it.

### 3.5. Quality Evaluation System Construction

To eliminate the influence of different units and data dimensions, we standardized the raw data for each quality index (Appendix A). Four principal functional components were constructed using the feature vector as the weight. We calculated the comprehensive principal component model by taking the ratio of the four principal components to the corresponding feature values and obtained the sum of the feature values of all extracted principal components as follows:F_1_ = 0.079*X*_1_ + 0.060*X*_2_ − 0.029*X*_3_ + 0.079*X*_4_ + 0.064*X*_5_ + 0.066*X*_6_ + 0.074*X*_7_ − 0.002*X*_8_ + 0.041*X*_9_ − 0.081*X*_10_
− 0.030*X*_11_ − 0.033*X*_12_ + 0.065*X*_13_ + 0.078*X*_14_ + 0.003*X*_15_ + 0.078*X*_16_ − 0.041*X*_17_ + 0.021*X*_18_ − 0.074*X*_19_ − 0.080*X*_20_
− 0.076*X*_21_ − 0.061*X*_22_
F_2_ = −0.017*X*_1_ − 0.085*X*_2_ + 0.150*X*_3_ + 0.034*X*_4_ + 0.113*X*_5_ − 0.104*X*_6_ + 0.019*X*_7_ + 0.173*X*_8_ − 0.151*X*_9_ + 0.002*X*_10_
+ 0.073*X*_11_ − 0.015*X*_12_ + 0.103*X*_13_ − 0.025*X*_14_ − 0.179*X*_15_ + 0.048*X*_16_ + 0.030*X*_17_ + 0.097*X*_18_ + 0.003*X*_19_ − 0.032*X*_20_ + 0.045*X*_21_ − 0.112*X*_22_
F_3_ = 0.081*X*_1_ + 0.183*X*_2_ + 0.000*X*_3_ + 0.001*X*_4_ + 0.012*X*_5_ − 0.011*X*_6_ + 0.109*X*_7_ − 0.059*X*_8_ + 0.040*X*_9_ − 0.026*X*_10_
+ 0.307*X*_11_ + 0.314*X*_12_ − 0.065*X*_13_ − 0.078*X*_14_ + 0.081*X*_15_ + 0.004*X*_16_ + 0.124*X*_17_ + 0.294*X*_18_ − 0.007*X*_19_ + 0.011*X*_20_
− 0.006*X*_21_ + 0.012*X*_22_
F_4_ = 0.025*X*_1_ + 0.021*X*_2_ − 0.304*X*_3_ − 0.030*X*_4_ − 0.010*X*_5_ + 0.094*X*_6_ + 0.173*X*_7_ + 0.200*X*_8_ − 0.176*X*_9_ + 0.021*X*_10_ − 0.084*X*_11_ + 0.209*X*_12_ + 0.059*X*_13_ + 0.009*X*_14_ − 0.083*X*_15_ − 0.016*X*_16_ − 0.508*X*_17_ + 0.106*X*_18_ + 0.271*X*_19_ − 0.003*X*_20_ + 0.143*X*_21_ + 0.161*X*_22_
F = 0.56147F_1_ + 0.24623F_2_ + 0.12282F_3_ + 0.06948F_4_

The comprehensive scores were calculated using the quality evaluation system, and the results showed that foxtail millet produced in Dingxiang (0.89) had the highest comprehensive quality, followed by Zezhou (0.10), Qinxian (0.02), and Xingxian (−0.12); the quality of foxtail millet in Yuci (−0.88) was the lowest (Table 2).

The quality of foxtail millet is affected by genetic traits [3], production area [22], environmental factors [21], harvest period [4,14], cultivation methods [41], and farming systems [42]. Similarly, variations in foxtail millet quality in different production areas were the result of comprehensive factors including variations in soil texture, fertility level, cultivation system, and climatic factors such as sunshine duration, temperature, and precipitation [21,29]. These factors can cause quality changes within the same foxtail millet variety. Therefore, the quality of Jingu 21 produced in the different areas showed substantial differences.

## 4. Conclusions

The appearance, nutritional value, culinary quality, and amino acid patterns of foxtail millet were significantly different in the different production areas. Strong correlations were observed among these quality indicators, which were classified into six groups with an average distance of 0.6. We established a quality evaluation system for foxtail millet that can objectively evaluate its comprehensive quality. The results of this evaluation showed that the comprehensive quality of foxtail millet, ranked from highest to lowest, was: Dingxiang > Zezhou > Qinxian > Xingxian > Yuci. Based on the results of this study, the foxtail millet value chain can be upgraded by ensuring the comprehensive quality of foxtail millet. In addition, the evaluation system can be also used for comprehensive quality scoring of foxtail millet from other areas and varieties.

## Figures and Tables

**Figure 1 foods-12-02545-f001:**
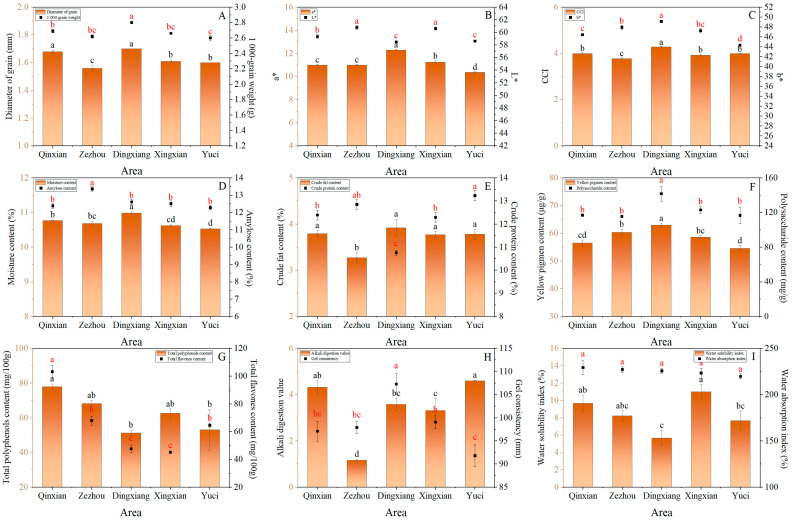
Appearance, nutritional value, and culinary quality indicator levels of foxtail millet “Jingu 21” at five production areas. Black and red letters indicate the significant differences at *p* ≤ 0.05 among the different treatments in bar chart and scatter chart.

**Figure 2 foods-12-02545-f002:**
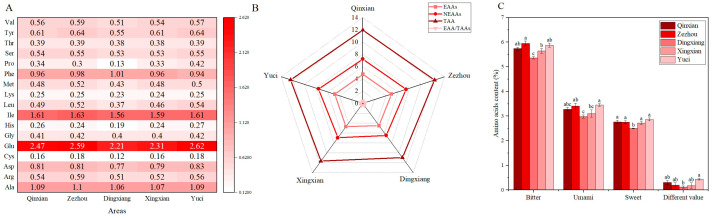
Amino acid pattern of foxtail millet “Jingu 21” at five production areas. (**A**) Heat map of amino acid composition and content of “Jingu 21”. (**B**) Radar map of essential amino acid (EAAs), nonessential amino acid (NEAAs), total amino acid (TAA) contents, and essential amino acid/nonessential amino acid (EAA/TAAs) of “Jingu 21”. (**C**) Flavor amino acid content of “Jingu 21”. Different letters indicate the significant differences at *p* ≤ 0.05 among the different treatments.

**Figure 3 foods-12-02545-f003:**
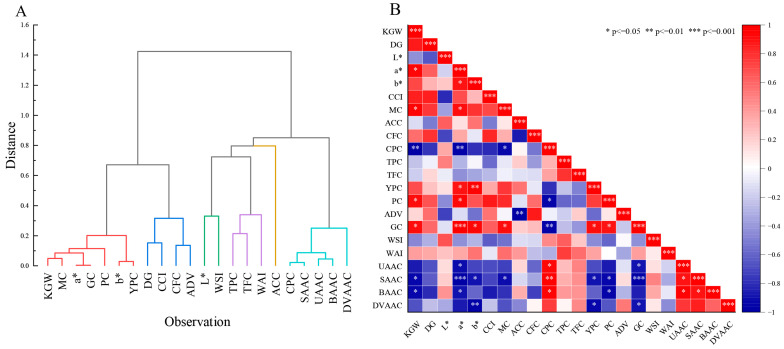
Clustering analysis and correlation analysis of foxtail millet “Jingu 21” at five production areas. (**A**) Hierarchical clustering analysis of 22 quality indicators. (**B**) The correlation between 22 quality parameters of foxtail millet. *, **, and *** indicate significance at 0.05, 0.01, and 0.001, respectively.

**Table 1 foods-12-02545-t001:** Eigenvalues of correlation matrix and eigenvectors of corresponding matrices for foxtail millet “Jingu 21” quality traits.

Primary Component		Principal Component Number			
		1	2	3	4
Eigenvalue		12.352	5.417	2.702	1.529
Percentage of variance (%)		56.147	24.623	12.282	6.948
Cumulative (%)		56.147	80.770	93.052	100.000
Load factor	KGW (X_1_)	0.971	−0.094	0.218	0.038
	DG (X_2_)	0.737	−0.460	0.495	0.032
	L* (X_3_)	−0.352	0.813	0.001	−0.464
	a* (X_4_)	0.982	0.184	0.003	−0.045
	b* (X_5_)	0.789	0.613	0.031	−0.015
	CCI (X_6_)	0.812	−0.564	−0.029	0.144
	MC (X_7_)	0.913	0.102	0.294	0.264
	ACC (X_8_)	−0.025	0.939	−0.158	0.305
	CFC (X_9_)	0.500	−0.816	0.108	−0.268
	CPC (X_10_)	−0.997	0.008	−0.070	0.032
	TPC (X_11_)	−0.374	0.393	0.830	−0.129
	TFC (X_12_)	−0.413	−0.082	0.849	0.319
	YPC (X_13_)	0.805	0.559	−0.176	0.090
	PC (X_14_)	0.968	−0.137	−0.210	0.014
	ADV (X_15_)	0.033	−0.967	0.218	−0.126
	GC (X_16_)	0.965	0.260	0.010	−0.024
	WSI (X_17_)	−0.509	0.160	0.336	−0.776
	WAI (X_18_)	0.263	0.523	0.795	0.162
	UAAC (X_19_)	−0.910	0.019	−0.018	0.415
	SAAC (X_20_)	−0.985	−0.171	0.031	−0.005
	BAAC (X_21_)	−0.944	0.245	−0.015	0.218
	DVAAC (X_22_)	−0.757	−0.605	0.033	0.246

**Table 2 foods-12-02545-t002:** Principal component score for foxtail millet “Jingu 21” quality traits.

	F1	F2	F3	F4	F	Ranking
Qinxian	−0.21	−0.33	1.74	0.05	0.02	3
Zezhou	−0.53	1.54	−0.26	0.70	0.10	2
Dingxiang	1.68	−0.22	−0.31	0.49	0.89	1
Xingxian	−0.01	0.20	−0.37	−1.74	−0.12	4
Yuci	−0.92	−1.20	−0.81	0.50	−0.88	5

## Data Availability

The data presented in this study are available on request from the corresponding author.

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
