# Peer review of "Development of a Comprehensive Quality Evaluation System for Foxtail Millet from Different Ecological Regions"

_foods, 2023, doi:10.3390/foods12132545_

Round 1

Reviewer 1 Report

The manuscript submitted for review is valuable and provides a new approach to the comprehensive qualitative classification of foxtail millet.

However, I have a few questions regarding the presented research results and their interpretation

1. Can we draw such far-reaching conclusions on the quality classification of grain based on only 5 cultivation places, one variety, and one harvest year?

2. In the experimental design chapter there is little information about agrotechnical procedures during foxtail millet cultivation

3. What type of resistant starch does foxtail millet contain (RS1, RS2, RS5?). Raw millet is not eaten, and heat-resistant starch can become digestible.

4. The content of bioactive substances was determined, why was the content of dietary fiber not determined?

5. Description of the content of flavor amino acids: umami, bitter,  sweet, should be found in the materials and methods section.

6. Figure 1 and 2 - graphs are not legible. Please increase the drawings' resolution and the axis descriptions' font (fig 1).

7. Figure 1 - cultivars are not continuous variables, you cannot connect points on the graph (KGW, L*, b*, ACC, etc.)

8. Please justify why 0.6 was taken as the cut-off point for cluster analysis (there is only one parameter in group 5…)

Author Response

请参阅附件

Reviewer 2 Report

Dear Authors,

Original research involves methods or results that have not been previously published. In your previous article entitled "The relationship between ecological factors and commercial quality of high-quality foxtail millet Jingu 21”", 12 locations were used but in the current manuscript only 5 locations were studied. Why!?
Best regards  

Reviewer 3 Report

Line 41: not for public… but for consumers. It is more suitable.

In material and methods sections, provide the measurements in different sub-items.

The amino acids were measured by near infrared, using the calibrations provided by supplier or by own calibration. How many samples were used to develop the models, please provided the values of models of calibration or validation. It is important to highlight that the results from near infrared are predictive values.

Improve the quality of figures, are not clear

In the legend of figure 2B provide the full name of abbreviations used in graph.

For Figure 2C how was measure the amino acids flavor? Using trained panel? How they are sure that flavors is due the amino acids amounts? Please, explain and add that information in manuscript.

Minor editing of English language required.

Round 2

Reviewer 1 Report

Dear Authors,

Thank you for responding to my review.

The manuscript may be published in this version. Unfortunately, the font charts are still too small in my opinion

Author Response

Thanks to the reviewer's suggestion, we have enlarged the font size in the figure again.

Reviewer 2 Report

Dear authors,

Please add your explanation in introduction section and or method section.

Best regards

Author Response

Thanks for the reviewer's suggestion, we have added the explanation content in Introduction and Materials and methods in lines 77-79 and 96-98.

Reviewer 3 Report

Add the values of calibration and validation methods of NIR provide by manufacture. 

Author Response

Thanks for the reviewer's suggestion. We have added this section to Materials and Methods, in lines 171-173.